# Dynamic Response of a Bridge–Embankment Transition with Emphasis on the Coupled Train–Track–Subgrade System

**Ping Hu [1,\*], Chunshun Zhang [2,\*], Wei Guo [3,\*] and Yonghe Wang [3]**

[1] School of Engineering Management, Hunan University of Finance and Economics, Changsha 410000, Hunan, China
[2] Department of Civil Engineering, Monash University, Clayton 3800, VIC, Australia
[3] School of Civil Engineering, Central South University, Changsha 410075, Hunan, China; yhwang45@csu.edu.cn
\* Correspondence: huping@hufe.edu.cn (P.H.); Ivan.Zhang@monash.edu (C.Z.); guowei@csu.edu.cn (W.G.)

**Abstract:** Dynamic response of a bridge–embankment transition is determined by, and therefore an indicator of, the coupled train–track–subgrade system. This study aims to investigate the approach of coupling the train–track–subgrade system to determine the dynamic response of the transition. The coupled system is established numerically based on the weak energy variation, the overall Lagrange format of D'Alembert's principle and dynamics of the multi-rigid body, which is verified by in-site measurements. With this model, the influence of rail bending, differential settlement and other factors on the dynamic performance of the transition system is analyzed. The results show that when the train driving speed is 350 km/h, basic requirements should be satisfied. These requirements include that the irregularity bending of the bridge–embankment transition section should be less than 1/1000, the rigidity ratio should be controlled within 1:6, and the length of the transition section should be more than 25 m. In addition, the differential settlement should not exceed 5 mm. Among these factors, the differential settlement and the bending of the rail surface are the main ones to cause the severe dynamic irregularity of the transition section. Our analysis also indicates a requirement to strengthen the 18 m and 25–30 m distance from the abutment tail and the bed structure.

**Keywords:** bridge–embankment transition section; bending of rail surface; differential settlement; dynamic performance

## 1. Introduction

The transition section, which connects soil subgrade and rigid structures (bridges, tunnels and culverts, etc.), is one of the weakest structures in the long-term operation of a high-speed railway [1]. Studying dynamic characteristics of the system is of great theoretical significance for the long-term stability and safe operation of the existing high-speed railway, and also provides technical reserves for future high-speed railway construction [2].

A field test is a direct way to study the dynamic characteristics of the transition section. However, it is expensive, labor intensive, and it is still challenging to cover various conditions. Alternatively, numerical simulation methods have been used to overcome the disadvantages of the field test method to study the dynamic characteristics of the transition section under different conditions [3]. There are three kinds of models:

(1) Train–track coupling model: This is mainly used to study the influence of the stiffness and deformation on the dynamic response and comfort of the train by simplifying the subgrade as a mass layer, spring and damper, even without considering the vibration effect of the subgrade. This is a

straightforward method adopted by many scholars. For example, this simple coupling model was employed to study the response of a track when the train ran on a transition zone [4]. The similar methodology was also used to establish a simplified three-dimentional (3D) train–track model to conduct sensitivity analysis on the vehicle speed, vehicle load, auxiliary track number, and track stiffness [5]. More recently, the dynamic responses of track in the marine environment were also modelled by the train–track coupling model [6].

(2) Track–subgrade coupling model: This is mainly used to study the characteristics of subgrade filling and has made a significant contribution to the research of subgrade dynamic characteristics by simplifying the trainload as moving load [7]. For example, the coupling model was adopted to study the dynamic response of the culvert embankment transition section, and also the effect of strengthening the foundation on reducing the uneven settlement of the transition zone was analyzed [8]. In addition, this coupling model could be combined with wave analysis technology to show that the gradual change of the stiffness of the transition section is conducive to the dynamic stability of track subgrade, as confirmed both by [9,10].

(3) Vehicle–track–subgrade model: The interaction between train, track and subgrade is considered in this type of model. For example, a bridge–embankment transition model was established by [11], where the vehicle was simplified as a multi-body system, and the vehicle–track–subgrade system was coupled by the wheel–rail contact force. The wheel–rail contact force serves an essential feature to couple the vehicle, track and subgrade. The similar methodology relying on the contact force was also found in [12,13], where [12] focused on the effect of filling materials on the dynamic properties of transition zones and [13] on the influence of bridge–pier settlement on the high-speed railway's dynamic response. In addition to the use of the wheel–rail contact force, the vehicle, track and subgrade could also be coupled using a fastener connection that couples the vehicle–track and track–subgrade subsystems, as illustrated in [14].

The above overview shows that the first and second modeling approaches oversimplify or ignore the dynamic response of either the subgrade or trains running on the track. While the third modeling approach acknowledges all contribution from the vehicle, track and subgrade, the modeling of the transition structure and material composition becomes very complex and a large number of system freedom degrees are generated. Therefore, the calculation efficiency is not satisfactory, and the complex modeling is not convenient for practical engineering application.

The authors have also been studying the dynamic responses of the transition section in recent years. On the one hand, in reference [15], we have carried out dynamic field tests under 120 high-speed vehicle running tests at speeds of 5–360 km/h and analyzed the governing factors of dynamic responses such as train speed, train running direction, vehicle axle load, and adjacent load. On the other hand, a 3D vertical coupling dynamic simulation method based on the D'Alembert's principle of energy weak variation and the Lagrange format were verified by the field test [16]. Further, a tunnel–culvert–tunnel transition section was established, which explored a more economical filling to replace the graded gravel +5% of the transition section. However, our above analysis in [16] simulated the action of a train by moving load but did not include the vehicle vibration that is critical to our current study.

To conclude, the existing numerical model has the following shortcomings: (1) The simplified model does not consider the vibration of subgrade or train, leading to some limitations for application; (2) The train–track–subgrade system of the transition section will generate a large number of system degrees of freedom for the complexity of the system, resulting in the low efficiency in analyzing. To overcome the above shortcomings, we first established the train–track coupling model based on the coupling theory; then the generated results from the coupled train–track model were used as a boundary condition of the other track–subgrade coupling model that is based on D'Alembert's principle. In doing so, it helped reduce the number of total freedoms of the coupling system and improved the efficiency when analyzing the dynamic responses of the track and subgrade. Furthermore, the influence of different influencing factors on the dynamic response of the bridge–embankment transition section system was studied deeply.

The paper is organized as follows: The general situation of the construction site of the transition section is described in Section 2. The numerical analysis model method of the transition section of the bridge–embankment is then described. After that, the numerical model considering the interaction and vibration of vehicles, track and subgrade are established and verified in Section 3. In Section 4, the change characteristics of vehicles, track and subgrade in the transition section along the longitudinal direction and the depth direction are analyzed in order to study the influence factors of the dynamic response of the bridge–embankment transition section. All the above analysis leads to the conclusions in Section 5.

## 2. Profile of the Bridge–Embankment Transition

By 2020, China's high-speed rail operating mileage has reached 35,000 km (Figure 1a), of which Wuhan–Guangzhou high-speed railway (Figure 1b) is the first high-speed railway with a speed of 350 per hour. The site in this paper is a filled bridge–embankment transition in the Wuhan–Guangzhou high-speed railway (HSR), covering from DK1252+840 to DK1252+888.27, with a total length of 48.27 m and the embankment height of about 3.5 m. The longitudinal aspect of the entire test section is shown in Figure 1c. In this section, the foundation is reinforced by cement fly-ash grave (CFG) piles with a pile diameter of 0.5 m; the transition section is with a triangle shape. The top of the pile is paved with 0.6 m-thick gravel cushion, and a layer of geogrid with the ultimate tensile strength not less than 80 kN/m is paved inside.

Formation lithology and geological structure: the covering layer of the hillock is eluvium and diluvia of Quaternary Holocene ($Q_4^{el+dl}$), locally mixed with crushed gravel and iron-manganese nodule with a thickness of 3–8 m, appearing brownish yellow or brownish red, which is in a hard-plastic state. The surface layer of the valley is silty clay with a thickness of 0–4 m, appearing brownish yellow, which is also in a hard-plastic state. The underlying Lower Jurassic ($J_S$) argillaceous siltstone is mixed with siltstone, locally mixed with the conglomerate, appearing purplish red, which is weak to completely weathered.

Hydrogeological conditions: The groundwater of mound is not developed, with a buried depth of 3.5–6 m, while the valley surface water is slightly developed, and groundwater with a buried depth of 0.1–1.6 m is locally developed. It contains a small amount of pore phreatic water, which is supplied by atmospheric precipitation. The surface water has weak corrosion of sulfuric acid type and medium corrosion of dissolution type to concrete, the underground water has weak corrosion of sulfuric acid type and weak corrosion of dissolution type to concrete.

The layout and data processing of the dynamic field measurement are shown in reference [15].

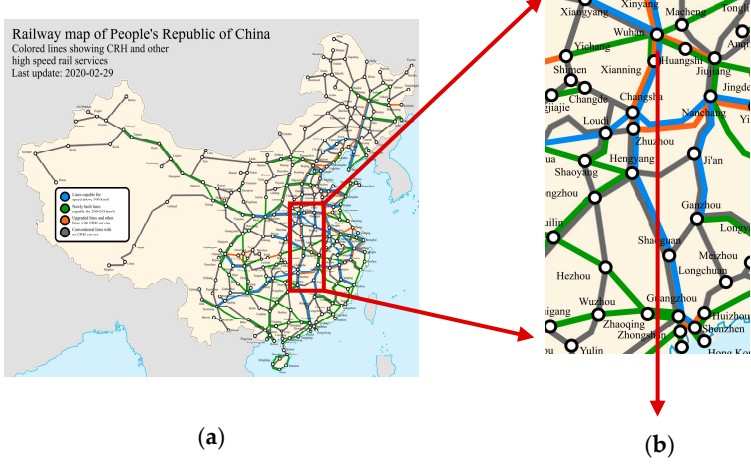

(**a**)　　　　　　　　　　　　　　　　(**b**)

**Figure 1.** *Cont.*

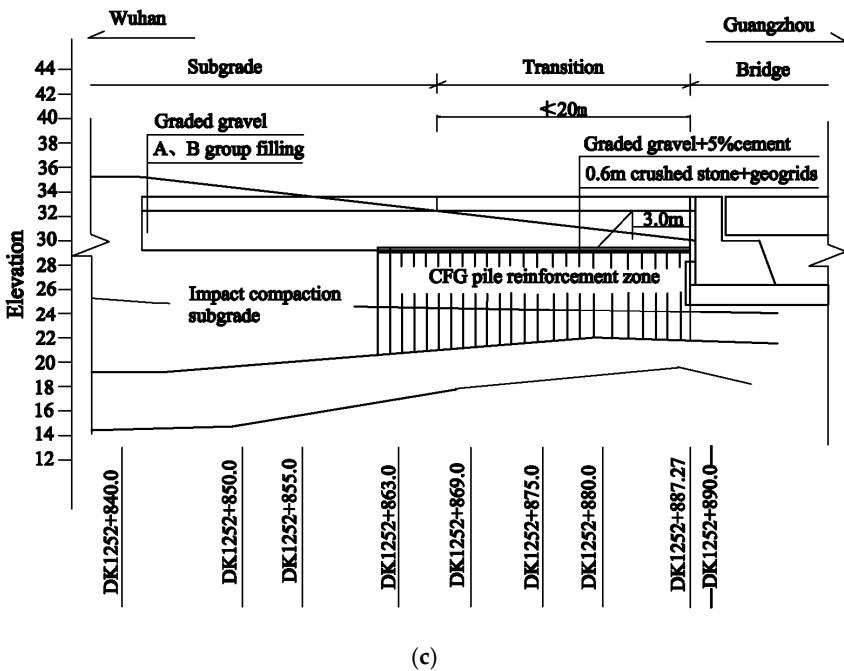

(**c**)

**Figure 1.** Profile of bridge–embankment transition section (unit: m): (**a**) China's medium- and long-term railway network planning (https://upload.wikimedia.org/); (**b**) a bridge transition section of Wuhan–Guangzhou high-speed railway; (**c**) a road bridge transition section in Xianning.

## 3. Establishment of Numerical Analysis Model of Bridge–Embankment Transition Section

### 3.1. Vehicle Model

High-speed trains are composed of locomotive vehicles and passenger vehicles. Various types of vehicle bogies have been designed according to different speed grades in the world. The fast or high-speed vehicle bogies are "three-no-structures" (no bolster, no shaking table and no side bearing). However, they are composed of two wheelsets, frame and secondary suspension from the perspective of energetics. The vehicle body, bogie and wheelsets are often regarded as rigid bodies, for example, the position of the vehicle body in space can be determined by three mutually perpendicular coordinate axes x, y and z of the center of gravity at point O with 6 degrees of freedom, such as stretching, traversing, moving up and down, side rolling, nodding and head shaking (as shown in Figure 2).

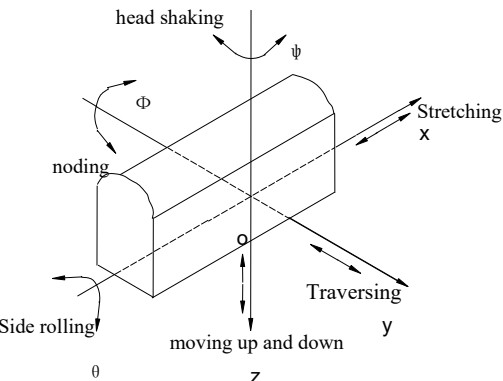

**Figure 2.** Spatial degrees of freedom of vehicle body.

The dynamic model of the passenger–track space coupling is shown in Figure 3. In Figure 3, $M_c$, $M_t$ and $M_w$ are, respectively, the mass of vehicle body, frame and wheelset. $I_{cxx}$, $I_{cyy}$ and $I_{czz}$

are, respectively, the inertia of rolling, nodding and shaking of vehicle body. $I_{txx}$, $I_{tyy}$ and $I_{tzz}$ are, respectively, the inertia of rolling, nodding and shaking of the frame. $I_{wxx}$, $I_{wyy}$ and $I_{wzz}$ are, respectively, the inertia of rolling, nodding and shaking of the frame. $m_r$ is the mass of unit length rail, and $EI$ is the bending rigidity of the rail. $M_s$ and $M_b$ are, respectively, the mass of the sleeper and ballast block. $K_{sx}$, $K_{sy}$ and $K_{sz}$ are, respectively, the longitudinal, transverse and vertical stiffnesses of the suspension on one side of the bogie. $C_{sx}$, $C_{sy}$ and $C_{sz}$ are, respectively, the longitudinal, transverse and vertical damping of the suspension on one side of the bogie. $C_{sdx}$ is the damping of an anti-hunting shock absorber. $K_{rx}$ is the anti-rolling stiffness. $K_{sx}$, $K_{sy}$ and $K_{sz}$ are, respectively, the longitudinal, transverse and vertical stiffnesses of the suspension on each axle box. $C_{sx}$, $C_{sy}$ and $C_{sz}$ are, respectively, the longitudinal, transverse and vertical damping of the suspension on each axle box. $K_{pv}$ and $K_{ph}$ are the vertical and horizontal stiffnesses provided by the rubber pad and fastener under the rail, while $C_{pv}$ and $C_{ph}$ are the vertical and horizontal damping provided by the rubber pad and fastener under the rail. Furthermore, $K_{bv}$, $K_{bh}$ and $K_{bw}$ are the vertical, horizontal and shear stiffnesses of the ballast, while $C_{bv}$, $C_{bh}$ and $C_{bw}$ are the vertical, horizontal and shear stiffness of the ballast, and $K_{fv}$ is the vertical stiffness of the subgrade and $C_{fv}$ is the vertical damping of the subgrade.

The structural difference between locomotive and passenger is that the former bogie is equipped with traction motor. For traction motor, there are three standard suspension modes: axle suspension, frame suspension and body suspension. Theoretically, each traction motor can be regarded as a separate rigid body, and there are significant differences between locomotive models with different motor suspension modes. However, the previous research shows that the vibration characteristics of the motor have little influence on the dynamic performance of the whole locomotive system, and the vibration of the motor can be ignored in the study of the whole locomotive system dynamics. Therefore, motor vibration is not considered in locomotive modeling.

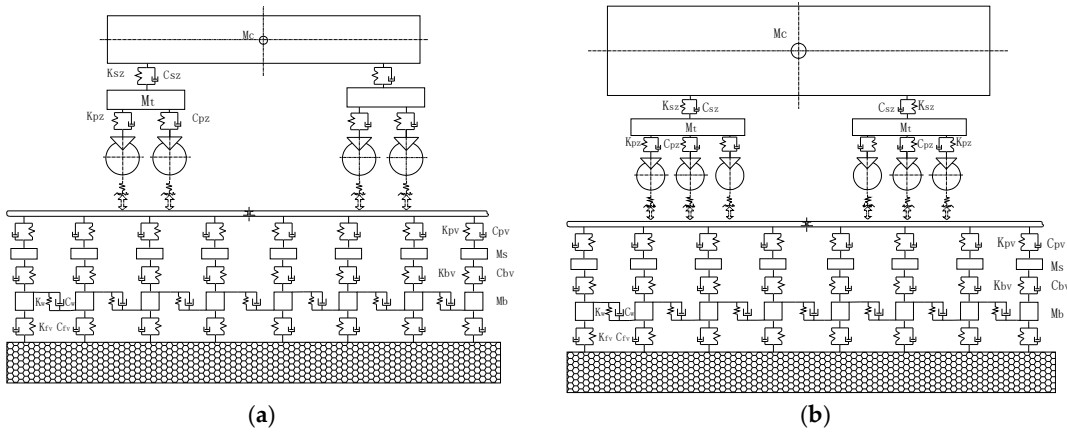

(a)  (b)

**Figure 3.** Typical vehicle–track space coupling dynamics model. (**a**) Passenger; (**b**) locomotive.

### 3.2. Coupled Vehicle–Track–Space Time-Varying Model

Each component of the vehicle is connected by springs and dampers, and each component is simplified as a force element or displacement element (as shown in Figure 4). The integrated model of slab track with a flexible foundation is shown in Figure 5, where the railway is composed of rail, slab, bed and subgrade. As shown in Figure 5, the rail is regarded as an infinite Timoshenko beam supported by sleepers that are modelled as springs and dampers. The slab is considered to be a rigid body, transferring the trainload to the embankment that is also modelled as springs and dampers.

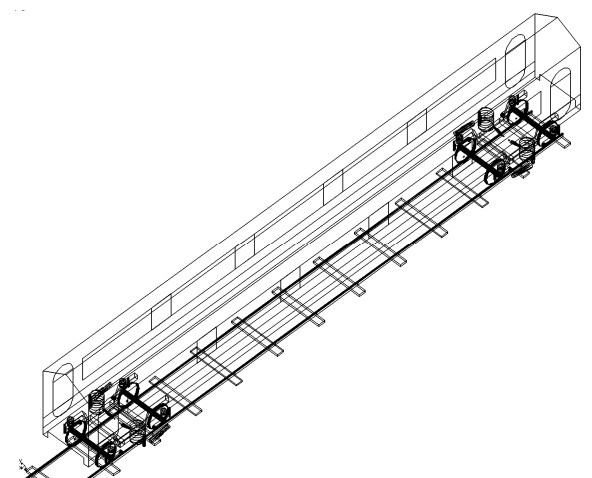

**Figure 4.** Integrated model of vehicle–track.

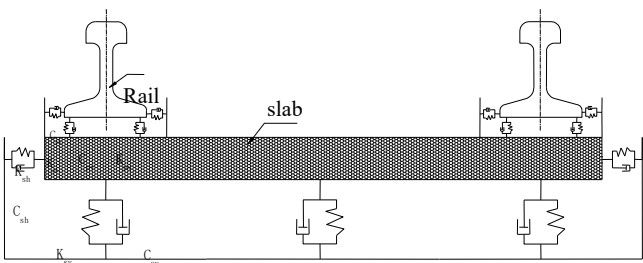

**Figure 5.** Integrated model of slab track with flexible foundation.

### 3.3. Solution of Dynamics of Multi-Rigid Body System

Cartesian coordinates of the mass center of the rigid body and Euler angles reflecting the orientation of rigid bodies are used as generalized coordinates. That is $q_i = \left[ x, y, z, \psi, \theta, \phi_i^T \right]$. Lagrange multiplier method is used to establish the multi-body dynamic equation:

$$\left. \begin{array}{c} \frac{d}{dt} \left( \frac{\partial T}{\partial \dot{q}} \right)^T - \left( \frac{\partial T}{\partial q} \right)^T + f_q^T \mu + g_{\dot{q}}^T \eta = Q \\ f(q,t) = 0 \\ g\left( q, \dot{q}, t \right) = 0 \end{array} \right\} \tag{1}$$

where $T$ is the kinetic energy of the system, $q$ is the generalized coordinate array of the system, $\dot{q}$ is the generalized velocity array of the system, $Q$ is the generalized force array, $\mu$ is the Laplace multiplier array of complete constraints, $\eta$ is the Laplace multiplier array of incomplete constraints, $f(q,t) = 0$ is the complete binding equation, $g\left( q, \dot{q}, t \right) = 0$ is the incomplete binding equation.

For a mechanical system with n degrees of freedom, the corresponding partial and angular velocities of each mass point and rigid body in the system, as well as the corresponding $N$ generalized active forces and generalized inertial forces can be calculated after determining $N$ generalized rates. Let the sum of the generalized active force and the generalized inertial force corresponding to each generalized rate be zero, then $N$ scalar equations, known as Kane equation, are obtained.

$$F^{(r)} + F^{*(r)} = 0 \ \ (r = 1, 2, \cdots, N) \tag{2}$$

where $F^{(r)}$ is the generalized active force, $F^{*(r)}$ is the generalized inertial force.

The matrix form of the above formula is:

$$F + F^* = 0 \tag{3}$$

Let $u = \dot{q}, \dot{u} = \ddot{q}$ in (1), therefore (1) turns into:

$$\left.\begin{array}{c} F\left(q, u, \dot{u}, \lambda, t\right) = 0 \\ G\left(u, \dot{q}\right) = u - \dot{q} = 0 \\ \Phi(q, t) = 0 \end{array}\right\} \tag{4}$$

where $u$ is the matrix of generalized velocity, $\lambda$ is the matrix of constraint reaction and force, $G$ is the matrix of algebraic equation describing constraint.

The gear algorithm is used to solve the motion equation of the system.

### 3.4. Finite Element Model of Track Subgrade of Road Bridge Transition Section

The condition of the transition section is described in Section 2. Based on reference [15], the finite element model is shown in Figure 6.

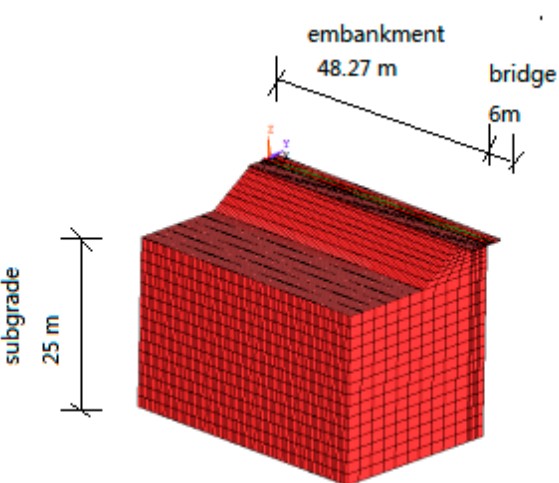

**Figure 6.** Finite element numerical model of track subgrade in road bridge transition section.

### 3.5. Load Application

Firstly, the vehicle–track model is established based on the dynamics of the multi-rigid body, and the relationship between vertical wheel load and time is obtained through simulation calculation. Then, this force is applied to the neutral axis of rail, i.e., the node of the beam element in the finite element model of rail and under rail structure, as shown in Figure 7. At any time $t$, the wheel load of the vehicle moving at a constant speed is:

$$f(t) = p \cdot \delta(x - vt) \tag{5}$$

$\delta(x_i = vt_i) = 1, \delta(x_i \neq vt_i) = 0, x_i = x_0 + i \cdot \ell_s (i = 1, 2, \ldots, n)$ is the position of the rail bearing groove of the track plate). In Equation (5), $p$ is the wheel load, $x$ is the position of the wheel load at the time $t$, $x_0$ is the initial reference point, $\ell_s$ is the rail bearing groove spacing 0.6 m, $n$ is the number of rail bearing grooves taken in the calculation model.

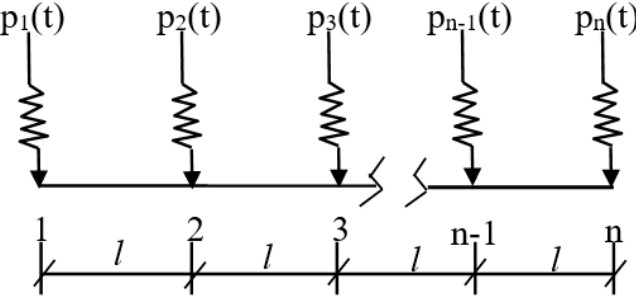

**Figure 7.** Schematic diagram of dynamic load application.

### 3.6. Model Validation

The field tests were carried out from DK1252+840 to DK1252+888.27 on the Wuhan–Guangzhou high-speed railway. The detailed description has been introduced in Section 2 and also reported in [15]. In order to validate the reliability of the transition section numerical model, the measured results of dynamic displacement and dynamic stress with those calculated by our simulation were compared.

The comparison is shown in Figure 8. It can be seen that there is a peak value between the simulated value of vertical dynamic stress and dynamic displacement and the test value in the longitudinal distance of the road bridge transition section, the simulated value of dynamic stress almost coincides with the test value, and the simulated value of dynamic displacement is slightly larger than the test value. However, all appear at about 24 m from the abutment tail. This result may be related to the parameter value of simulation calculation, the simplification of constraint conditions, and the particularity and difference of test conditions. However, in general, the difference between the simulation calculation value and the test value is not significant, and the trend is the same with the line change, so the finite element model established in the verification text is correct.

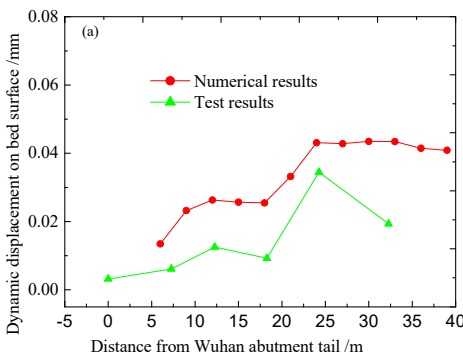
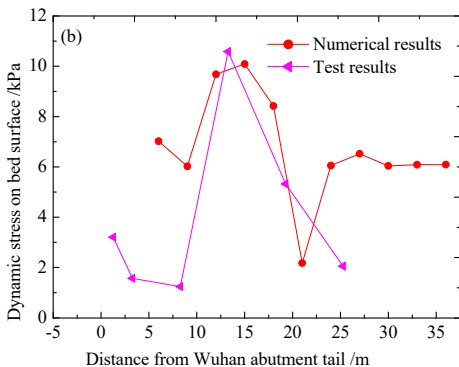

**Figure 8.** Comparison of measured and numerical results of dynamic stress amplitude on the surface of subgrade bed. (**a**) Dynamic displacement; (**b**) dynamic stress.

## 4. Influence Factors on Dynamic Responses of Vehicle-Track-Subgrade System in Transition Section

Wuhan–Guangzhou high-speed railway adopts a ballastless track, which requires stricter and smaller deformation of subgrade. In order to reveal the dynamic response law of the bridge–embankment transition system, the author analyzes the track irregularity characteristics under the conditions of transition angle, differential settlement, stiffness and train speed, and then studies the dynamic characteristics of the vehicle, track and subgrade along the longitudinal direction and depth direction under the action of the train.

### 4.1. Change Rule of Dynamic Response along the Longitudinal Direction of the Line

Based on the vehicle–track–subgrade coupling model of the transition section, the dynamic performance of vehicles under the above different track irregularity conditions is calculated, and the

results are shown in Tables 1 and 2. The change rules between differential settlement, bending angle change, stiffness ratio and length of transition section and the surface dynamic response of transition section subgrade bed along the line direction are shown in Figures 9–12.

**Table 1.** Comparison of dynamic performance of four kinds of vehicles with irregularities.

| Working Condition | | | Acceleration of Train Body (m/s2) | | Rate of Wheel Load Reduction Q/P | Y/Y$_{Derailment}$ | Rail Wheel Force (kN) | |
|---|---|---|---|---|---|---|---|---|
| | | | Lateral ay | Vertical az | | | Vertical | Lateral |
| Rigid | Transition length | 10 | – | 0.893 | 0.120 | 0.010 | 115.3 | – |
| | | 15 | | 0.582 | 0.110 | 0.010 | 115.8 | |
| | | 20 | – | 0.424 | 0.105 | 0.010 | 116.4 | – |
| | | 25 | – | 0.405 | 0.063 | 0.010 | 118.2 | – |
| | | 30 | 0.018 | 0.406 | 0.054 | 0.011 | 121.3 | – |
| | Bending angle * | 1/1000 | 0.033 | 1.035 | 0.124 | 0.015 | 123.8 | – |
| | | 3/1000 | −0.145 | 4.413 | 0.330 | 0.050 | 142.7 | – |
| | | 4/1000 | 0.152 | 4.327 | 0.522 | 0.069 | 150.2 | – |
| | PSD & | | 0.349 | 0.500 | 1.469 | 2.041 | 140.6 | 193.7 |
| Flexible | PSD | | 0.341 | 0.520 | 0.448 | 0.229 | 135.7 | 41.9 |
| | Bending angle of 0.75/1000 | | 0.153 | 0.733 | 0.142 | 0.215 | 117.7 | – |

Note: Y$_{Derailment}$ is the maximum allowable value of the lateral displacement between wheel and rail, which is −0.045 m according to the requirements of the German Railway code. Y/Y$_{Derailment}$ is the ratio of the lateral displacement between wheel and rail to the maximum allowable lateral displacement between wheel and rail. Derailment coefficient = Q/P, where Q is the lateral force, and P is the vertical force. * Bending angle indicates the ratio of differential settlement to transition length. & PSD (power spectral density) indicates the power spectral density of track irregularity.

**Table 2.** Influence of different differential settlements (mm) of road bridge transition sections on vehicle's dynamic performance.

| Differential Settlement | Acceleration of Train Body (m/s²) | | Rate of Wheel Load Reduction Q/P | Y/Y$_{Derailment}$ | Wheelset Vertical Displacement (mm) | Rail Wheel Force (kN) | |
|---|---|---|---|---|---|---|---|
| | Lateral a$_y$ | Lateral a$_y$ | | | | Vertical | Lateral |
| 3 | 0.07 | 0.45 | 0.53 | 0.19 | 40.4 | 2.9 | 226.6 |
| 4 | 0.12 | 0.67 | 0.79 | 0.25 | 57.7 | 3.7 | 238.0 |
| 5 | 0.12 | 0.87 | 0.96 | 0.33 | 87.5 | 7.3 | 283.2 |
| 6 | 0.19 | 3.49 | 2.88 | 1.46 | 139.2 | 9.4 | 294.2 |
| 7 | 0.26 | 3.79 | 4.37 | 1.66 | 168.6 | 15.4 | 344.9 |
| 7.4 | 0.50 | 4.93 | 8.48 | 2.21 | 183.3 | 56.1 | 558.5 |

It can be seen from Figure 9 that the maximum amplitude of dynamic stress and displacement on the bed surface does not vary significantly with the stiffness. However, the amplitude varies significantly in the longitudinal direction of the railway line, especially in the transition section. The higher the stiffness of subgrade is, the smaller the dynamic displacement on the bed surface is, and the higher the dynamic stress and the dynamic stress on the bed surface are. Table 1 shows that under the power spectral density (PSD) irregular conditions, the absolute rigid foundation is not suitable for high-speed vehicles. This is because the calculated dynamic response in the case of a rigid subgrade exceeds the relevant regulations, such as those specified in 95J01-L and 95J01-M. In comparison, for the flexible subgrade that meets the requirements of the high-speed railway ballastless track specification, the dynamic response obtained by the simulation calculation does not exceed the various prescribed values mentioned above. This shows that it is safe for high-speed vehicles to run on flexible subgrades under PSD irregular conditions.

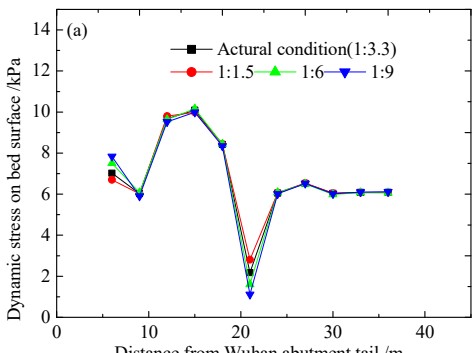 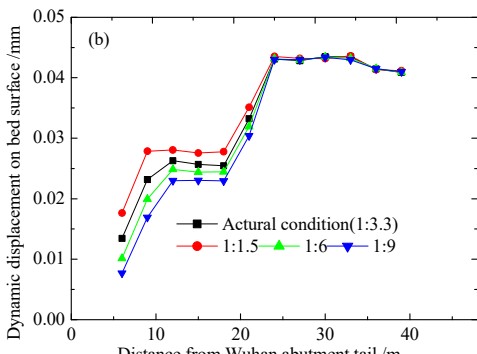

**Figure 9.** Effect of stiffness change on dynamic response of subgrade surface layer: (**a**) dynamic stress; (**b**) dynamic displacement.

The rigidity of the rail causes the irregularity of the rail surface, and the more serious the irregularity is, the larger the rail deflection difference is. The stiffness ratio of the transition section and common subgrade for the safety and stability of vehicle operation is not the main factor to control the dynamic characters of the transition section. On the other hand, the on-site investigation of the state of the road bridge transition section shows that as long as the track is frequently lifted and removed and the ballasted track is tamped for maintenance, the track surface is kept smooth, and the track inspection vehicle can pass smoothly without being deducted points.

It can be seen from Figure 10 that with the increase in the transition angle $\theta$, the amplitude of dynamic stress and displacement on the bed surface does not change significantly with the transition angle, but its amplitude changes differently in the longitudinal direction of the transition section. Table 1 shows that the vehicle lateral acceleration $a_y$, vertical acceleration $a_z$, wheel load reduction rate Q/P, Y/Y$_{Derawment}$, wheel pair vertical force, lateral wheel-rail force Q and so on increases in turn as the bending of the track surface increases. When the angle $\theta$ = 3/1000, the acceleration of the vehicle has exceeded the design limit of 0.13 g; while when $\theta$ = 1/1000, the acceleration of the vehicle is smaller than the design limit of 0.13 g. This confirms that the limit value of the angle of the ballastless track is 1/1000, which is reasonable. At the same time, the bending angle of the transition section has a more significant impact on the dynamic behavior of the vehicle body on the rail than the performance under the rail. As the dynamic performance changes such as vehicle acceleration and wheel load reduction rate are caused by track irregularity, after the length of transition section is determined, the dynamic performance of the vehicle body can be guaranteed as long as the bending angle is limited.

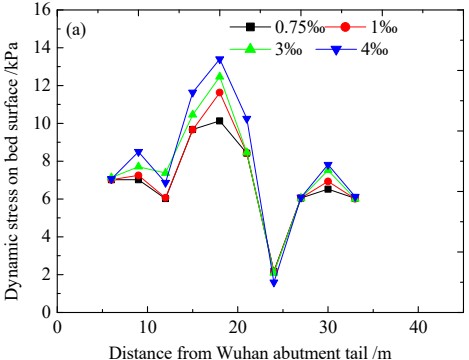 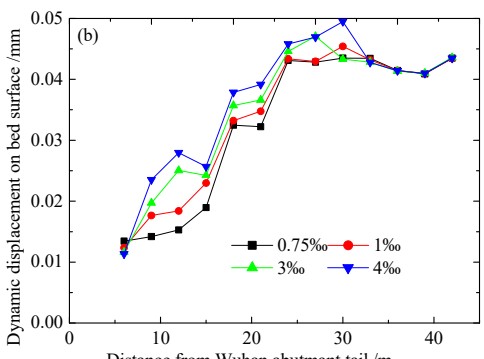

**Figure 10.** Effect of angle change on dynamic response of subgrade surface layer: (**a**) dynamic stress; (**b**) dynamic displacement.

It can be seen from Table 1 and Figure 11 that with the increase in differential settlement, the dynamic response of vehicle–track increases in turn. When the differential settlement h = 5 mm, most of the dynamic characteristics' evaluation indexes are less than the allowable values, and some indexes are close to the allowable values of the dynamic characteristics' control indexes, such as the wheel–rail load reduction rate, which reaches 0.96, and the vehicle body acceleration, which reaches 0.87 m/s$^2$. It can also be seen that the dynamic response of the transition section increases with the increase in differential settlement of the transition section. When the differential settlement of the transition section is greater than 5 mm, the dynamic response tends to increase significantly as the differential settlement increases. In addition, it is found that the differential settlement of the transition section makes the concave and convex of the curve of dynamic parameters along with the track trend more prominent, which will lead to more severe vibration in the process of the train operation. This also shows the influence of the differential settlement of the foundation on the dynamic parameters of the transition section, indicating the critical factors that must be considered in the design of the transition section.

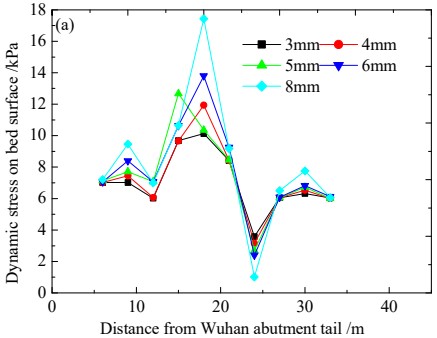 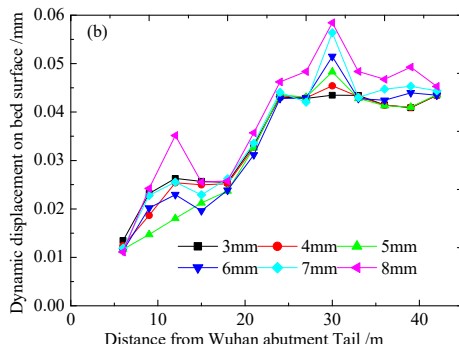

**Figure 11.** Effect of differential settlement on the dynamic response of the subgrade surface layer: (**a**) dynamic stress; (**b**) dynamic displacement.

According to the literature [1], the acceleration of the vehicle body is a primary evaluation standard to confirm the length of the transition section. It can be seen from Figure 12 that with the increase in the length of transition section, the dynamic stress and displacement amplitude on the bed subgrade changes little, and the dynamic response tends to be gentle along the longitudinal direction, especially when the length of the transition section is longer than 25 m. Table 1 shows that when the length of the transition section is 8 m and 10 m, the vertical acceleration of the vehicle centroid decreases sharply with the increase in the transition section length. When the length of the transition section is more than 20 m, variance of vertical acceleration of the vehicle centroid is not apparent. When the length of the transition section is 10 m, the vertical acceleration of the vehicle centroid is more than 0.8 m/s$^2$, close to 1.0 m/s$^2$. The comprehensive analysis shows that when the train passes at a high speed of 350 km/h when the length of the transition section is more than 15–20 m, the changes of relevant dynamic indexes are minimal, and then the length of the transition section continues to increase, almost without any effect. The theoretical calculation results also show that even though the length of the transition section is as short as 10 m, or even 5 m, the vertical vibration acceleration of the vehicle body, wheel–rail vertical force and other indicators increase to a certain extent. However, the value is still at a relatively low level, far below the corresponding control value. Therefore, it is better to take 25 m as the transition section length of this construction site.

From the above analysis, it can be concluded that the maximum longitudinal dynamic stress of the bed surface reaches a peak at 18 m from the tail of the abutment, and the dynamic displacement increases with the distance from the abutment and reaches a peak at 25–30 m from the tail of the abutment.

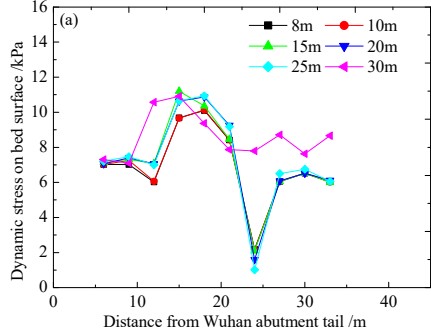 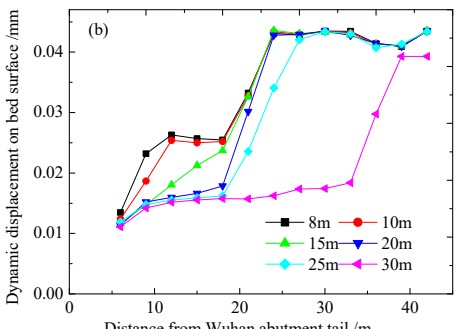

**Figure 12.** Effect of transition length on dynamic response of surface layer of subgrade bed: (**a**) dynamic stress; (**b**) dynamic displacement.

### 4.2. Law of Dynamic Response along the Depth

Figure 13 indicates the rule that dynamic response changes with depth when the transition section of the positive trapezoid and inverted trapezoid is adopted at Section 3 m from the abutment. It can be seen that the elastic strain firstly increases sharply with the increase in depth, which increases nearly five times from the surface of the bed surface to the bottom of the bed surface, increases nearly 1–2 times from the surface of the bottom bed to the bottom of the bed bottom, and then gradually decreases from the bottom of the bed bottom to the subgrade, as shown in Figure 13a. On the contrary, the dynamic stress decreases sharply with the increase in subgrade depth, which decreases by 44% from the surface of the bed surface to the bottom of the bed surface, decreases by 36.32% from the surface of the bed bottom to the bottom of the bed bottom, then the subgrade slowly decreases below the bottom of the subgrade bed, as shown in Figure 13b. Thus, we can infer that the bed layer bears the primary dynamic response, and the type of filling has a significant influence on the dynamic response in the transition section.

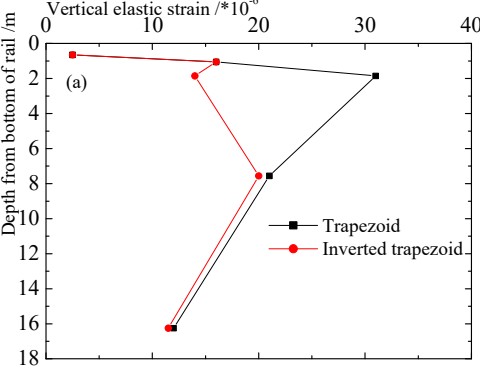 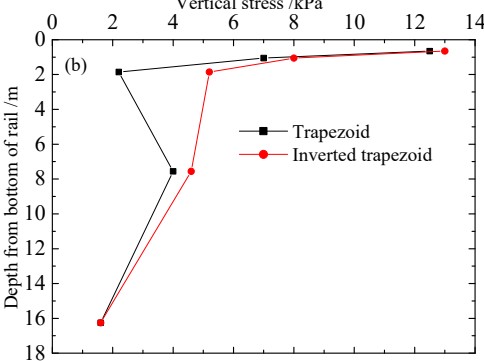

**Figure 13.** Variation of dynamic response with depth: (**a**) vertical elastic strain; (**b**) vertical dynamic stress.

The curves of dynamic stress and vertical elastic strain in Figure 13a,b indicate that the major difference only occurs within the layer of approximately 2–7.5 m (bed and shallow subgrade). In detail, from Figure 13a, for the inverted trapezoidal embankment, the vertical dynamic elastic strains of the bed bottom and the subgrade are smaller than those of the trapezoid. The response of dynamic stress is just the opposite as shown in Figure 13b, that is: the dynamic stress of the inverted trapezoidal embankment is greater. After that, the two curves gradually coincide when the depth exceeds 7.5 m. The above phenomena, due to the rigidity of the transition section filling, are higher than that of ordinary subgrade filling and the corresponding stress has a better diffusion effect. Therefore, the analysis result of this paper considers that the structural design of inverted trapezoid is more reasonable than that of the positive trapezoid transition section.

## 5. Conclusions

Many scholars, including the author, have been studying the dynamic characteristics of the transition section of high-speed railways for a long time, mainly focusing on the track–subgrade coupling system. The train action is normally simulated by moving load, measured acceleration and other methods, without considering its vibration function. As a result, this leads to inadequate research in vibration and interaction between the vehicle, rail and subgrade. To fully consider the vibration and interaction, a coupled train–track–subgrade system was established with train–track coupling theory and D'Alembert's principle. The proposed model was verified by in-site measurements and focused on the study of the influencing factors that dominate the dynamics of the coupled system. Specifically, the following conclusions were obtained.

(1) It is confirmed that differential settlement and rail bending angle in the transition section are the main factors that cause dynamic irregularity in the transition section. There are some other indirect controlling factors such as rigidity ratio and length of the transition section, which may reduce the differential settlement and rail bending, leading to less dynamic responses of the coupled system.

(2) Under the operation of a high-speed train with the speed of 350 km/h, the above four factors are recommended to meet the following range: the differential settlement shall not exceed 5 mm, the limit value of the rail bending angle of the bridge–embankment transition shall be less than 1/1000, the rigidity ratio shall be controlled within 1:6, and the length of the transition section shall be 25 m.

(3) With regard to satisfying the conditions in (2), our research found that stress and displacement have the following distribution characteristics. On the one hand, in the longitudinal direction of the roadbed, the maximum dynamic stress and dynamic displacement of the transition section appear between the rigid bridge and the flexible roadbed. For example, the dynamic stress peaks at the roadbed surface 18 m away from the tail of the abutment, while the dynamic displacement peaks at 25–30 m away from the tail. On the other hand, in the depth direction of the subgrade, the bed layer resists the main dynamic response, so the dynamic stress and dynamic displacement attenuate sharply in the depth direction. Based on the above analysis, in order to reduce the dynamic stress and dynamic displacement response of the transition section, it is recommended to reinforce the subgrade at a distance of 18 m and 25–30 m longitudinally from the abutment, and to strengthen the bed layer in the depth of the subgrade.

(4) This paper also preliminarily concludes that under the condition of power spectral density (PSD) of track irregularity, a flexible infrastructure is much safer than a rigid infrastructure. Nevertheless, under different working conditions, the changes of the train acceleration, wheel load reduction rate, track subgrade dynamic stress, dynamic displacement, and other indicators on the flexible and rigid subgrades need further study.

**Author Contributions:** Writing—original draft preparation, P.H.; writing—review and editing, C.Z. and W.G.; project administration, Y.W.; funding acquisition, P.H. and Y.W. All authors have read and agreed to the published version of the manuscript.

**Funding:** Financial support from Hunan Provincial Natural Science Foundation (No. 2020JJ5488) and National Natural Science Foundation of China (No. 50678177) are gratefully acknowledged.

**Conflicts of Interest:** The authors declare that they have no conflict of interest.

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
