# Peer review of "Dynamic Response of a Bridge–Embankment Transition with Emphasis on the Coupled Train–Track–Subgrade System"

_applsci, doi:10.3390/app10175982_

Round 1

Reviewer 1 Report

The topic is quite interesting and relevant as it covers the Applied Sciences’ aim and scope clearly. Though this article should be published in Applied Sciences journal, however, there are some things the authors need to work on before it is ready to be published. I have provided my insights as follows:

  1. I would advise the author to compile the citing references in texts, like on page no. 1, line 31 and 32, “culverts, etc.), is one of the weakest structures in the long-term operation of a high-speed railway [1][2][3][4]” should be “culverts, etc.), is one of the weakest structures in the long-term operation of a high-speed railway 32 [1-4]. This should be done through out the manuscript.
  2. Page 11, Line no. 321, could it be possible that the authors kindly check and correct the units such as 8 m/s2, the 2 on second should be superscript like 8 m/s2. Furthermore, kindly check through the manuscript on how you write your units and measurements uniformly, for example, Page 11, line 319, “….section is more than 20 m, the varies….”, and at other places it is “…roadbed surface 18m away from…” on Page 13, line 381.
  3. Some text in the manuscript is in red, is this anything special? If not then I would advise you to change it back to black colour like the rest of the manuscript.
  4. Please kindly name the all abbreviations, variables and constants, Page 6, line 180, the eq. 2 contains Fr + F*(r) = 0, I could not find what Fr and F*(r) were, until I tried to read above paragraph. However, for other variables and constants, you have mentioned what is what. Page 6, line 186, like G is the matrix of algebraic equation describing constraint.
  5. In Page 6, line 186, “G is the matrix of algebraic equation describing constraint, t, G is the matrix of algebraic equation describing generalized velocity.” is repeated, please remove one.
  6. Page 7, line 202, “In Eq (1), p is wheel load…” eq. 1 is on page 5, line 168 and that equation does not contain any p variable. I assume you are referring to eq. 5 on page 6, line 200 “f (t) = p ×d (x - vt)” if so please do correct the equation numbers you are referring to through out the manuscript.
  7. In figure 8, there is a mention of test results, which test results are being referred here? Did the authors do any experimental tests and used those to compare with the numerical results, if so could you provide more details.
  8. I would kindly ask the authors to ask native English speaking person to proofread this manuscript, as I have found some grammatical (past, present and future tenses) mistakes.

Author Response

Dear  Professor,

Thanks for your kind comments.

The detailed changes are as follow:

Comment Point 1:  I would advise the author to compile the citing references in texts, like on page no. 1, line 31 and 32, “culverts, etc.), is one of the weakest structures in the long-term operation of a high-speed railway [1][2][3][4]” should be “culverts, etc.), is one of the weakest structures in the long-term operation of a high-speed railway 32 [1-4]. This should be done through out the manuscript.

Authors’ Reply: Good point! Thanks! All references have been modified accordingly.

Comment Point 2: Page 11, Line no. 321, could it be possible that the authors kindly check and correct the units such as 8 m/s2, the 2 on second should be superscript like 8 m/s2. Furthermore, kindly check through the manuscript on how you write your units and measurements uniformly, for example, Page 11, line 319, “….section is more than 20 m, the varies….”, and at other places it is “…roadbed surface 18m away from…” on Page 13, line 381.

Authors’ Reply: Noted with thanks; all these unit issues have been revised. 

Comment Point 3Some text in the manuscript is in red, is this anything special? If not then I would advise you to change it back to black colour like the rest of the manuscript.

Authors’ Reply: Revised with thanks.

Comment Point 4: Please kindly name the all abbreviations, variables and constants, Page 6, line 180, the eq. 2 contains Fr + F*(r) = 0, I could not find what Fr and F*(r) were, until I tried to read above paragraph. However, for other variables and constants, you have mentioned what is what. Page 6, line 186, like G is the matrix of algebraic equation describing constraint.

Authors’ Reply: Thanks for pointing these out. The relevant explanation has been added in revised line 184.

Comment Point 5: In Page 6, line 186, “G is the matrix of algebraic equation describing constraint, t, G is the matrix of algebraic equation describing generalized velocity.” is repeated, please remove one.

Authors’ Reply: Thanks. We have removed it.

Comment Point 6: Page 7, line 202, “In Eq (1), p is wheel load…” eq. 1 is on page 5, line 168 and that equation does not contain any p variable. I assume you are referring to eq. 5 on page 6, line 200 “f (t) = p ×d (x - vt)” if so please do correct the equation numbers you are referring to through out the manuscript.

Authors’ Reply: Yes, it has been corrected to be eq (5) in the revised line 205.

Comment Point 7: In figure 8, there is a mention of test results, which test results are being referred here? Did the authors do any experimental tests and used those to compare with the numerical results, if so could you provide more details.

Authors’ Reply: Thanks. Clarification regarding the test results for comparison has been added from the revised line 216 to 221.

Comment Point 8: I would kindly ask the authors to ask native English speaking person to proofread this manuscript, as I have found some grammatical (past, present and future tenses) mistakes

Authors’ Reply: Thanks for your kind suggestion. The 2nd author, Dr Chunshun Zhang, who has been working in an Australian university for over ten years, has double-checked and corrected the grammatical issues. If the reviewer still finds a similar problem and informs us, we would be very grateful.

Thanks again!

Best Regards,

Ping

Reviewer 2 Report

This work must be cautiously re-edited. There are many missing spaces between the words. The same papers appear twice in the reference list:

[1] and [19], [4] and [17], [5] and [21], [8] and [16].

Some comments are given below:

  1. Introduction

Please split information concerning every single reference describing what is really interesting in referenced papers. If you write [1][2][3][4][5] it does not explain why this particular article is important for your research. And you use this form many times:  [6][7][8][9][10] and [11][12][13].

Referring to your own papers [21] and [22] please explain how your current work is (or is not) related to previous research.

  1. Profile of the bridge ...

Please provide a readable Figure 1c with information about dimensions and soil profile taken for the analysis.

  1. Establishment of numerical ...

Figure 7 needs some corrections: cut the lines between p3 and pn-1.

I'd suggest to split section 4 into separate sections for Results of numerical analysis and Discussion. But it is not mandatory.

Author Response

Dear Professor,

Thank you for your kind comments.

The detailed changes are as follow:

Comment Point 1: This work must be cautiously re-edited. There are many missing spaces between the words. The same papers appear twice in the reference list:

[1] and [19], [4] and [17], [5] and [21], [8] and [16].

 Authors’ Reply: Revised with thanks.

Comment Point 2: Please split information concerning every single reference describing what is really interesting in referenced papers. If you write [1][2][3][4][5] it does not explain why this particular article is important for your research. And you use this form many times:  [6][7][8][9][10] and [11][12][13].

Authors’ Reply: Thanks; we have assigned the related references accordingly.

Comment Point 3: Referring to your own papers [21] and [22] please explain how your current work is (or is not) related to previous research.

Authors’ Reply: Thanks. The clarification has been added in revised lines 62-64.

Point 4:  Please provide a readable Figure 1c with information about dimensions and soil profile taken for the analysis.

Authors’ Reply: Added with thanks.

Comment Point 5: Figure 7 needs some corrections: cut the lines between p3 and pn-1.

Authors’ Reply: Cut with thanks.

Comment Point 6:  I'd suggest to split section 4 into separate sections for Results of numerical analysis and Discussion. But it is not mandatory.

 Authors’ Reply: Thank you for your suggestion. Sections 4.1 and 4.2, respectively, simulated and analyzed two critical dynamic responses along the longitudinal and the depth. To highlight the difference and stress the importance of the above two aspects, we, therefore, adopted the 4.1 and 4.2, respectively. However, we are very grateful for the reviewer’s suggestion and will follow it if it applies to any other paper.

Thanks again!

Best regards,

Ping

Reviewer 3 Report

The paper explores the dynamic response of a bridge-embankment transition of the coupled train-track-subgrade system. The authors try to investigate the coupling of train-track-subgrade system and calculate the dynamic response of the transition. The coupled system was established numerically based on the energy weak variation and the overall Lagrange format of D’Alembert principle and dynamics of multi rigid body. The model verified by in-site measurements. The influence of rail bending, differential settlement and rigidity ratio on the dynamic performance of the transition system was calculated. According to the numerical results the authors state that when the train driving speed is 350 km/h, the irregularity bending of the transition section of the road and bridge should be less than 1/1000, and the rigidity ratio should be controlled within 1:6, the length of the transition section should be more than 25 m. Furthermore, the differential settlement should not exceed 5mm. Finally, the authors conclude that under the condition of power spectral density, (PSD) of track irregularity, flexible infrastructure is much safer than rigid infrastructure. Nevertheless, under different working conditions, the changes of the train acceleration, wheel load reduction rate, track subgrade dynamic stress, dynamic displacement and other indicators on the flexible and rigid subgrades need further study

The paper addresses a topic posing numerical and experimental challenges and having practical significance. It is methodologically correct. The paper is suitable for publication.

Some comments to improve the paper:

  • In line 66 the generalized coordinates should be written correctly as (x,y,z,ψ,θ,φ)Τ
  • The literature review in introduction is thorough and it is very well written, however some additional references listing below regarding to dynamic response of a bridge-embankment transition to curved bridges subjected to earthquake loading and standardization of bridge structures for different spans could be added in the introduction.
  1. Nikolaos Pnevmatikos, Vassilis Sentzas “Preliminary estimation of response of curved bridges subjected to earthquake loading.” Journal of Civil Engineering and Architecture, Volume 6, No. 11 (Serial No. 60), pp. 1530–1535, 2012, ISSN 1934-7359, USA.
  2. Gianis Mantzaris, Nikos Pnevmatikos, Gewrgia Tsiboukaki, Alekos Mantzaris, “Standarization of bridge structures for spans up to 100m”, Concrete Plant International journal, CPI, Volume 3, June 2010.

Author Response

Dear Professor,

Thank you for your kind comments!

The detailed changes are as follow:

Comment Point 1: In line 66 the generalized coordinates should be written correctly as (x,y,z,ψ,θ,φ)Τ

 Authors’ Reply: Revised with thanks in revised line 169.

Comment Point 2: The literature review in introduction is thorough and it is very well written, however some additional references listing below regarding to dynamic response of a bridge-embankment transition to curved bridges subjected to earthquake loading and standardization of bridge structures for different spans could be added in the introduction.

1.Nikolaos Pnevmatikos, Vassilis Sentzas “Preliminary estimation of response of curved bridges subjected to earthquake loading.” Journal of Civil Engineering and Architecture, Volume 6, No. 11 (Serial No. 60), pp. 1530–1535, 2012, ISSN 1934-7359, USA.

2.Gianis Mantzaris, Nikos Pnevmatikos, Gewrgia Tsiboukaki, Alekos Mantzaris, “Standarization of bridge structures for spans up to 100m”, Concrete Plant International journal, CPI, Volume 3, June 2010.

Authors’ Reply: Added with thanks.

Thanks again!

Best regards,

Ping

Round 2

Reviewer 2 Report

Dear Authors

I appreciate the amount of work that was done to refer to reviewer's comments. Most of my suggestions were accepted, some were just discussed (I respect your right to support your original position).

I can still see major faults concerning format of the references (authors' and paper's descriptions).

I cannot see any justification for new references [10] and [11]. Reference [10] is about numerical calculations of curved concrete bridges. Paper [11] is about standardization of bridge structures. It has nothing to do with your text: "A field test is a direct way to study the dynamic characteristics of the transition section [5-7]. However, it is expensive and labour-intensively, and It is still challenging to cover various conditions [9-11]."

Please read carefully any paper that you want to add to the reference list and provide clear information: what valuable information for your study could be derived from those papers.

You partially accepted my previous comment about splitting of the references but still, even the cautious reader cannot find why you refer individually to many of them.

Please check again carefully the final pdf file because converting from word file may lose some spaces between words and change font formats.

Author Response

Dear Professor,

Thank you very much for giving us an opportunity to revise our manuscript.

We have studied your comments carefully and have made revision which marked in yellow in the paper. We have tried our best to revise our manuscript according to your comments. Attached please find the revised version, which we would like to submit for your kind consideration.

We would like to express our great appreciation to you for comments on our paper.

Thank you and best regards.

Yours sincerely,

Ping
